# Experimental Research on Uniaxial Compression Constitutive Model of Hybrid Fiber-Reinforced Cementitious Composites

**DOI:** 10.3390/ma12152370

**Published:** 2019-07-25

**Authors:** Tao Cui, Haoxiang He, Weiming Yan

**Affiliations:** Beijing Key Laboratory of Earthquake Engineering and Structural Retrofit, Beijing University of Technology, Beijing 100124, China

**Keywords:** hybrid fiber concrete, SIR model, constitutive model, elasticity modulus

## Abstract

In order to establish accurate compressive constitutive model of Hybrid Fiber-Reinforced Concrete (HFRC), 10 groups of HFRC specimens containing polyvinyl alcohol (PVA), polypropylene (PP), and steel fibers are designed and compressive testing is conducted. On the basis of summarizing and comparing the existing research, accuracy of various stress-strain constitutive model is compared and the method of calculating fitting parameters is put forward, peak stress, peak strain, and elastic modulus of specimens with different fiber proportion are analyzed, the calculation expressions of each fitting parameter are given. The results show that, under the condition that the volume of the hybrid fiber is 2% with the proportion of the steel fiber increase, the strength of the specimen increases, the peak strain decreases slightly, and the elastic modulus increases significantly. In specimens mixed with PVA-PP hybrid fiber, with the increase of PVA fiber proportion, the peak stress and elastic modulus of the material are improved, and the peak strain are decreased. The existing stress-strain expressions agree well with the tests. Accuracy of exponential model proposed in this paper is the highest, which can be applied in engineering and nonlinear finite element analysis of components.

## 1. Introduction

In recent years, Fiber-reinforced Concrete (FRC) is widely used in civil engineering. It is well-known for the improvement of crack arrest mechanism of traditional concrete due to the chaotic distribution of fibers in the cement matrix: the friction and slip between the hybrid fiber can prevent the development of crack and inhibit damage accumulating [1,2]. Fibers currently used in FRC include polyvinyl alcohol fiber (PVA), polypropylene fiber (PP), and steel fiber. The incorporation of single kind of fiber can only improve the performance of a certain aspect of concrete. For example, steel fiber can increase the tensile and compressive bearing capacity, but is easily corroded and may reduce fluidity of concrete. PVA fiber and PP fiber can significantly improve tensile capacity and deformability of concrete, but elastic modulus usually decreases [3,4]. For solving the problem above, Hybrid Fiber-reinforced Concrete (HFRC) combining two or more kinds of fibers is created. Researches have shown that HFRC has excellent properties such as tensile and compressive strength, toughness, and durability at an appropriate dosage [5,6,7,8,9].

At present, substantial theoretical and experimental researches on HFRC have been conducted. For example, Qian and Stroven [10] pointed out that PP-teel hybrid fiber have significant toughening effect on concrete, and can also increase concrete ultimate compressive strain, the cracking strength of HFRC and steel fiber-reinforced concrete (SFRC) is almost the same, but after the peak load the descending section is flatter than that of SFRC. Mei and Xu [11] pointed out that steel fiber plays a role in improving compressive strength, flexural strength, and fracture energy in hybrid fibers. PP fiber can improve the toughening effect and make the material strain-hardened. Rambo and Silva [12] proved that polyolefin (PO) fiber and steel fiber can significantly improve the ductility, fracture energy, and residual strength of the specimen; the effect of hybrid fibers is higher than respective single mix due to steel fibers optimizes the placement of the PO fibers in the matrix, thereby enhancing the toughness of the material.

Compressive constitutive relationship is a necessary condition of HFRC component and construction research. Its accuracy could significantly affect the design results of structures applying this material. However, at present, most design codes neglect the role of fibers in the compression process of fiber-reinforced concrete, which is considered for the purpose of conservative in structural design. In fact, if only use the constitutive relation of ordinary concrete, it is difficult to reflect the stress-strain law of fiber-reinforced concrete, and the application of this model to structural analysis will cause deviation. At the same time, the different kinds and contents of fibers in FRC will also affect the stress-strain relationship of the specimens under compression. Therefore, in scientific research, it is of great theoretical and practical significance to adopt more refined models to characterize the stress-strain relationship of the specimens and to better reflect the differences of the amount and types of fibers. Although numerous researches have been conducted on the toughening effect and mechanical property of HFRC, the commonly used HFRC compression constitutive model is mainly obtained by modifying the parameters in existed concrete constitutive model and the theoretical depth is insufficient. Research on effects of each kind of fiber on compressive constitutive model is also rare. To some extent, these shortcomings restrict the accuracy of material performance simulation, structural performance analysis, and application in engineering. In view of this, this paper designed 10 sets of uniaxial compression test and 10 sets of cubic compression test. Under a constant volume fraction of fibers, the mechanical characteristics of cubic compressive strength, axial compressive strength, peak strain, and elasticity modulus under different kind and proportions of fibers are analyzed. The relationship between characteristics above and fiber type and content is also given through experiments. In addition, this paper also summarizes the expression of stress-strain relationship of concrete materials appearing in the current literature. After optimizing its parameters, the accuracy of the expression is compared in the rising and falling stages. Calculation results can provide reference for engineering and nonlinear finite element analysis of the material components and structures. It is worth pointing out that this study is based on the macro-scale, that is, calibrate the parameters of material constitutive relations (such as empirical model, elastic-plastic model, damage constitutive relationship, etc.) based on macro-scale experiment, and then apply them to the numerical model to calculate the mechanical behavior and structural properties, ignoring the mesostructural characteristics, for example, microcracks and inclusions; this kind of research has a small amount of calculation and is widely used in engineering, but it cannot reveal the intrinsic law of material damage evolution. At present, in the field of advanced damage mechanics, some scholars have put forward some advanced theories at the micro-level, such as discrete modeling of fibers and matrix lattices [13], phase field model [14], and gradient damage model [15]. The experiment in this paper can also provide data support for further theoretical research.

## 2. Experimental Program

### 2.1. Materials and Mix Proportions

Nine groups of HFRC specimens with 2% volume fraction of fibers and 1 group of matrix specimens are designed with different ratios of PVA, PP, and steel fibers. Cubic compression test and axial compression tests were carried out. Among them, 3 specimens are in each group, totaling 30 cubic specimens and 30 prism specimens. Side length of cubic specimen is 150 mm, size of prism specimen is 150 mm × 150 mm × 300 mm. Mix proportions of each group are listed in Table 1, water-binder ratio (w/b) is 0.35, Ordinary Portland cement P.O 42.5 is applied as binder of HFRC, according to the information provided by the manufacturer, its chemical composition is listed in Table 2, quartz sand with fineness modulus 2.69 is selected as fine aggregate, and the polycarboxylic acid water reducer with reducing rate of ~20% is adopted. The PVA fiber applied is produced by Kuraray Co., Ltd. (Tokyo, Japan); equivalent diameter is 39 μm and length is 12 mm, the ultimate tensile strength is no less than 1200 MPa. Steel fibers are corrugated fibers with length of 25 mm and equivalent diameter is 0.4 mm. The ultimate tensile strength is not less than 1100 MPa. Polypropylene (PP) fibers applied are made in China with length of 12 mm and equivalent diameter of 31 μm. The ultimate tensile strength is not less than 500 MPa.

### 2.2. Specimen Preparation and Maintenance

HFRC is a kind of multiphase composite material, and some defects would inevitably occur in the process of specimen forming and curing, such as microcracks on the interface between cement matrix and aggregate. Strict quality control measures have been taken to minimize these defects and the damage caused to the specimens during the manufacture of the specimens, and to reduce the dispersion of the test results. The process of specimen preparation and maintenance is as follows.

(1)Cement, fly ash, and quartz sand are added in turn to the mixer according to the calculated mix proportion and dry mix for 2 min. Hybrid fibers are added slowly during the dry mixing process, ensuring that the process is even and slow.(2)After the dry mixing is completed, water and water reducer are added according to the mixing ratio and stirred in the mixer for 5 min.(3)After stirring, the HFRC slurry is poured out and loaded into the template. The process of loading into the template is as fast and uniform as possible.(4)Place the template on the shaking table and vibrate evenly. In order to prevent the segregation of concrete, the vibration time is ~30 s.(5)The specimens are demolded 24 h and cured in the standard curing room. The temperature is kept at 20 °C, the relative humidity was 95%, and the curing period was 28 days.

The specimens are loaded by MTS press system with maximum loading capacity of 1000 kN. The whole loading process is controlled by deformation velocity; the loading velocity is 0.05 mm/min. In order to measure the strain accurately, the longitudinal deformation is measured by displacement sensor and strain gauge. The strain gauge is arranged on the opposite sides of the specimen along the loading direction. The loading device is shown in Figure 1. The common hydraulic press releases excessive deformation energy at the moment of material failure, and collected stress-strain curve only has ascending section. In order to obtain the complete curve, two jacks with the maximum loading capacity of 500 kN are installed on the loading plate of the press. They bear the reaction force of the loading plate at the failure moment, consuming the deformation energy released by pressing head and prevent the specimen from instantaneous failure and sudden loss of bearing capacity.

## 3. Test Results and Analysis

### 3.1. Failure Process and Failure Patterns

Because of evident crack resistance effect, fibers crossing cracks and near crack tips transfer stress to the upper and lower surfaces of cracks: the stress concentration at crack tips decreases, the failure process is slower, the descending section of strain-stress curve is gentler, and the toughness of specimens is improved. Figure 2 is normalization stress-strain curve of PC2, where *f*_cf_ is the peak stress and *ε*_f_ is the peak strain. Taking PC2 as an example, axial compression process of HFRC can be divided into the following stages.

(1) Elastic stage (OA, *σ* < 0.3*f*_cf_).

When *σ* ≤ 0.3*f*_c_, the major deformation of specimens is elastic deformation, the initial microcracks have not developed; the crack resistance of fibers has not been fully exerted; the strain-stress curve is similar to that of plain concrete, rising linearly; and a few concave sections exists in the curve, which is due to tiny gaps between the loading plate and specimen.

(2) Stable crack development stage (AB, 0.3*f*_cf_ < *σ* < 0.85*f*_cf_).

With the increase of stress, numbers of microcracks appear on the specimen surface and extend from loading ends to inside of specimen. However, the cracks are fine and the cracking process is slow due to the crack resistance of fibers, the slope of stress-strain curve does not vary significantly before cracking and no obvious macrocracks exist on the surface of specimen.

(3) Unstable crack development stage (BC, 0.85*f*_cf_ < *σ* < *f*_cf_).

When stress increases to about 0.85*f*_cf_, the crack development of the specimen is unstable, macrocracks appeared. Transverse expansion of specimen began to accelerate. The fibers spanning the cracks still play the role of crack resistance, which reduces the crack growth rate. Hence, the peak strain of the specimen increases. When the stress-strain curve reaches point C, the cracks in the specimen connect and form a failure surface, and the macrocracks on the surface are obvious. The fibers across the failure surface are pulled out from the matrix, and a small amount of fiber is broken. The bearing capacity of the specimens reaches its peak value.

(4) Decline stage (CDEF).

When the specimen reaches the peak stress, it loses bearing capacity rapidly and the curve drops suddenly. When it reaches point D, a series of vertical macrocracks parallel to the loading direction appear on specimen surface, and then the rate of stress decline slows down. Therefore, point D can be defined as an inflection point. As shown in Figure 3b. When the stress-strain curve decreases to point E, the vertical crack penetrates. Although bridging fibers of the failure surface are invalid, fibers spanning cracks along the direction of crack development can still prevent its development. Through stress redistribution, the cracking section is closely distributed, and the failure part can transmit part of the pressure, so the specimens can still bear part of the load when deformation is large. Then, the main crack of the specimen develops continuously, forming a failure zone, but bearing capacity does not decrease significantly, a platform with slow stress decline appears. The corresponding stress value of the platform is defined as residual stress. The residual stress of the specimens is approximately 20% to 30% of the peak stress. For group PC, the higher the content of PVA fiber is, the higher the residual stress is. For group SC, the higher the content of steel fiber is, the higher the residual stress is.

For matrix specimens, several main cracks are noticeable when destroyed. The crack penetration divides the specimen into several small prisms and the bearing capacity decreases rapidly. The test has not collected a complete descending section curve. In specimens on PC group, the main cracks are the fine longitudinal cracks when the specimens are destroyed. By observing the cracks, it can be observed that the fibers in the middle of the interface are pulled out. The main failure mode of specimens in SC group is oblique shear crack. It can also be observed that PP–steel hybrid fibers are pulled out between the fracture surfaces. The failure modes of typical specimens in each group are presented in Figure 3.

A large number of existing constitutive model curves of hybrid fiber-reinforced concrete under compression also have the above stages, but due to the difference of fiber mix ratio, the shapes of each stage are slightly different. For example, in CHI’s research [16], due to the low content of polypropylene fibers and the steep descent stage, the polypropylene fibers content in this paper is high. In this test, besides the crack resistance, we also expect that polypropylene fibers can play a more toughening role after cracking, but the strength and elasticity modulus of PP fibers are low, which will reduce the strength and elasticity modulus of matrix. At the same time, the reduction of peak stress and elasticity modulus is more obvious.

### 3.2. Stress-Strain Curve under Uniaxial Compression

In order to obtain accurate test data, the average stress of specimens in one group with the same strain value is taken as the corresponding stress at the strain value, and by this way, the stress-strain curve of each group of specimens can be obtained.

Similar to plain concrete, the stress-strain curves of HFRC specimens contain ascending and descending sections. In order to decrease the error, the rising part of the curve is measured by the strain gauge pasted on the surface of the specimens, and the falling part is measured by the displacement sensor. The complete stress-strain curve of specimens is shown in Figure 4.

It can be observed from Figure 4 that regular occurrence of ascending sections is relatively uniform. In the specimens mixed with PVA-PP hybrid fibers, the peak stress decreases by 5.1 to 12.1% compared with the matrix, however the peak strain increases by 2.5 to 4.4 times with the PVA fibers content increase, elastic modulus of the specimens decreases and the peak strain increases. For specimens mixed with PP-steel hybrid fibers, with the increase of the steel fiber content, the peak stress and elastic modulus of the specimens increased significantly and peak strain decreased. The descending section of the stress-strain curve is quite different: compared with the matrix DB, the strain of the specimens mixed with hybrid fiber increases obviously in descending section after the peak point, and the residual stress is almost the same. The ultimate strain of the specimens is over three times larger than the matrix. The compressive deformation ability of the specimens is improved significantly. From the analysis above, it can be seen that in hybrid fiber system, steel fibers can increase the compressive strength, PVA and PP fibers will reduce the strength and elastic modulus of specimens, but will improve the ductility and deformation ability of specimens. The PVA fibers applied in this experiment have higher strength and elastic modulus, therefore the strength and elasticity modulus of HFRC specimens are less affected.

### 3.3. Peak Stress and Peak Strain

The measured peak stress and its error diagram are shown in Figure 5.

Based on the measured data and exist research [16,17,18], the formulas for calculating peak stress are presented as
(1)$$f_{c}=f_{c0}(1+\alpha_{\mathrm{pf}}\lambda_{\mathrm{pf}}+\alpha_{\mathrm{sf}}\lambda_{\mathrm{sf}}+\alpha_{\mathrm{vf}}\lambda_{\mathrm{vf}})$$
in which *f*_sp_ and *f*_vp_ represent the peak stress of PP-steel hybrid fiber and PVA-PP hybrid fiber specimens, respectively; *f*_c0_ represents the peak stress of the matrix; and *α*_sf_, *α*_pf_, and *α*_vf_, are the peak stress fitting parameters related to the content and aspect ratio of steel fiber, PP fiber, and PVA fiber, respectively. *λ*_sf_, *λ*_pf_, and *λ*_vf_ represent the factors of steel fiber, PP fiber, and PVA fiber, respectively. They are the product of fiber content and aspect ratio. It is noteworthy that when the fitting parameter of a certain fiber is 0, it means that the content of the fiber is 0. The fitting results are *α*_sf_ = 0.2706, *α*_pf_ = −0.02542, *α*_vf_ = −0.01409. The test and fitting results are listed in Table 2. The fitting correlation coefficient *R*^2^ = 0.901 and the fitting effect is satisfied. In this paper, the peak strain of specimens is calculated considering the effect of hybrid fibers. According to the mechanical mixing law of composite materials [13,14,15], the peak strain of two kinds of specimens can be obtained, and the peak strain can be expressed as
(2)$$\varepsilon_{c}=\varepsilon_{c0}(1+\beta_{\mathrm{sf}}\lambda_{\mathrm{sf}}+\beta_{\mathrm{pf}}\lambda_{\mathrm{pf}}+\beta_{\mathrm{vf}}\lambda_{\mathrm{vf}})$$
where *ε*_fsp0_ and *ε*_fvp0_ are the peak strain of steel-PP and PVA-PP hybrid fiber specimens, respectively, and *ε*_c0_ is the peak strain. *β*_sf_, *β*_pf_, and *β*_vf_ are the peak strain fitting parameters related to the content of steel fiber, PP fiber, and PVA fiber, respectively. Based on test data in existing research [13,14,15,16], it can be obtained by fitting analysis that *β*_sf_ = 0.4751, *β*_pf_ = 0.1666, and *β*_vf_ = 0.2121. The experimental and fitting results are listed in Table 2. The test data of existed researches are compared with the calculation results of Equations (1) and (2). The calculation results are shown in Figure 6. It is worth pointing out that although the volume fraction of PP fibers in the specimens of PP-steel hybrid fiber-reinforced concrete used in reference [14] is less than 0.15%, and the volume fraction of steel fibers varies from 0.5% to 1.5%; factors such as fiber type and aspect ratio are taken into account, as far as the factor of fiber content is concerned, the test results in this paper are based on the fiber blending scale. The present law of change is in good agreement with Equations (1) and (2). Based on Chi’s research [16], the range of fiber content is enlarged, and the calculation results can still be expressed by the above two formulas. In Wang’s research [17], PVA-steel hybrid fibers are used, but the above formulas can also be used for calculation. It can be seen that for hybrid fibers system, the effect of the hybrid fibers on peak stress and strain can be approximately identified as the linear superposition of the components.

### 3.4. Elastic Modulus

It can be observed that the elastic modulus of concrete is not enhanced significantly by mixing of steel fibers, and even decreases due to the increase of more interfacial weak areas in the interior of the concrete. The formulas for calculating the elastic modulus of this material are suggested as
(3)$$E_{\mathrm{sp}}=\frac{10^{5}}{2.2+(34.7/f_{\mathrm{fcu},\mathrm{sp}})}(1-\gamma_{\mathrm{sf}}\lambda_{\mathrm{sf}}-\gamma_{\mathrm{pf}}\lambda_{\mathrm{pf}})$$
(4)$$E_{\mathrm{vp}}=\frac{10^{5}}{2.2+(34.7/f_{\mathrm{fcu},\mathrm{vp}})}(1-\gamma_{\mathrm{vf}}\lambda_{\mathrm{vf}}-\gamma_{\mathrm{pf}}\lambda_{\mathrm{pf}})$$
in which *E*_sp_ and *E*_vp_ are the elastic modulus of PP-steel hybrid fiber and PVA-PP hybrid fiber specimens, respectively; *f*_fcu,sp_ and *f*_fcu,vp_ are cubic compressive strength of specimens mixed with PP-steel hybrid fiber and PVA-PP hybrid fiber, respectively; and *λ*_sf_, *λ*_pf_, and *λ*_vf_ are the characteristic values of steel fiber, PP fiber, and PVA fiber, respectively, which are the product of fiber content and aspect ratio. *γ*_sf_, *γ*_pf_, and *γ*_vf_ are fitting parameters related to the content of different kind of fibers. After regression analysis of test results of this test and exist research, this research suggests *γ*_sf_ = 0.3078, *γ*_pf_ = 0.0777, and *γ*_vf_ = 0.1443. The calculated and measured results are shown in Table 3. The predicted results conducted by this paper and existed experiments are shown in Figure 7. The difference between calculated and measured values is small. The elastic modulus of the specimens can be obtained by Equations (3) and (4). It is worth pointing out that the above formulas are applicable when the volume fraction of fibers is 0–2%. Further research is needed on the applicability of the formulas when the volume fraction of fibers is higher, and the shape of steel fibers will also affect the pressure characteristics. However, more experimental studies are needed to quantify this effect.

## 4. Stress-Strain Curve under Uniaxial Compression

### 4.1. Introduction of SIR Model

In this article, a load-deformation constitutive model of HFRC is established by introducing SIR model of infectious disease transmission dynamics [19]. The SIR model divides the population in the epidemic area into three categories: the susceptible group S, referring to those who have not been infected; the infected group, I, referring to those who have been infected with infectious diseases, can spread to the members of class, S; and the Removed group, R, referring to those who have been isolated or immunized by the disease. Thus, when the total number remains unchanged, the population can be divided into three categories: susceptible, infected, and Removal, with the time variable *t*. The proportion of these three groups can be recorded as *S*(*t*), *I*(*t*) and *R*(*t*), and *S*(*t*) + *I*(*t*) + *R*(*t*) = 1. The SIR model can be expressed as the following nonlinear differential equation by assuming that the average number of effective contacts per patient per day is *θ*, the proportion of patients cured or removed per day to the total number of patients is *μ*, and the increase rate of susceptible persons is *η*, when the number of patients increases. The speed parameter of reducing patients after taking preventive measures is *ρ*. Therefore, the SIR model can be expressed as follows.
(5)$$\left\{ \begin{array}{l} \frac{di}{dt}=\theta is-\mu i\\ \frac{ds}{dt}=-\theta is+\eta i-\rho i^{2}s \end{array}\right.$$


From analysis above, it can be observed that the SIR model reflects the dynamic evolutionary process of internal structure and self-characteristics of a system with different components under external action, and has certain universality.

Continuous damage and failure of HFRC under uniaxial compressive load will occur. Three kinds of elements exist in any state: undeformed element, deformed element and fracture, or removed element. Stress-strain relationship of HFRC is a direct reflection of this dynamic evolution. Hence, similar performance parameters of HFRC can be compared with the SIR model. Strain *ε* reflects the deformation ability and process of HFRC, corresponding to time variable *t* of the SIR model. *s*(*ε*) is defined as the equivalent stress of the element that has not yet deformed, *σ*(*ε*) is the equivalent stress of the element in the deformed state, and *r*(*ε*) is the equivalent stress of the element in the fracture or withdrawal state, therefore, *s*(*ε*) + *σ*(*ε*) + *r*(*ε*) = 1. *θ* can be expressed as unit stress transfer rate, *μ* can be expressed as unit failure rate, *η* is the increase rate of force element when failure element increases, and *ρ* is the speed parameter that reduces failure element after stress transfer and distribution. Based on the failure mechanism of HFRC under uniaxial compression, the stress-strain model reflects the dynamic changes of HFRC in the elastic stage. The elastic-plastic stage and plastic stage can be established as follows.
(6)$$\left\{ \begin{array}{l} \frac{d\sigma }{d\varepsilon }=\theta \sigma s-\mu \sigma \\ \frac{ds}{d\varepsilon }=-\theta \sigma +\eta \sigma -\rho \sigma^{2}s \end{array}\right.$$

Equation (6) is a unique solution for a system of nonlinear differential equations, but it cannot be solved analytically by mathematical methods. Generally, it can only be solved by numerical methods. In this paper, based on the homotopy analysis method non-zero auxiliary parameters and non-zero auxiliary functions with high convergence rate are introduced. Finally, the analytical expression of Equation (6) is obtained as follows.
(7)$$\left\{ \begin{array}{l} \sigma \left(\varepsilon \right)=\sum_{m=1}^{+\infty }\sum_{k=1}^{3m+2}\delta_{m,k}e^{-k\kappa \varepsilon }\\ s\left(\varepsilon \right)=s\left(0\right)+\sum_{m=1}^{+\infty }\sum_{k=1}^{3m+2}\eta_{m,k}e^{-k\kappa \varepsilon } \end{array}\right.$$
where, *k* = *μ* − *θs* (∞) ≈ *μ*, *m*, *θ*, *η*, and *k* are all coefficients. According to the characteristics of the above analytical solution, the approximate analytical solution of Equation (7) is expressed as
(8)$$\sigma =\sum_{i=1}^{n}{\left(-1\right)}^{i+1}c_{i}e^{-k_{i}\kappa \varepsilon }$$

On this basis, if *n* = 2, the stress-strain constitutive model of HFRC in the form of exponential function with natural constants as the bottom can be obtained.
(9)$$\sigma =c\left(e^{b\varepsilon }-e^{a\varepsilon }\right)f$$


In fact, Japanese scholar Umemura has proposed a kind of concrete constitutive relation in exponential which is similar to Equation (9) under the condition of *n* = 2, but the constitutive relation is determined by experiment data and cannot be explained well by existed theory. The solution presented in this paper has strict theoretical basis. Results of the two equations can be verified mutually.

### 4.2. Forms of Stress-Strain Models

Most researchers use piecewise function to fit the stress-strain curve of concrete. The function forms are summarized in Table 4. It can be seen from the table that the ascending section can be generally divided into polynomial, exponential, trigonometric function and rational fraction, and other forms, but the descending section is currently widely used in a single form, most of which use rational fraction.

The stress-strain curves measured in this paper are analyzed by nonlinear regression according to the suggested formula in Table 4. The fitting parameters of ascending curve can be calculated. The results are shown in Figure 5. It can be observed from the figure that various fitting formulas can fit the ascending curves of HFRC well. The fitting results of rational fraction and exponential form are slightly better than those of polynomial form. For polynomial fitting function, with the increase of parameter *A*, the ascending section of curve becomes steeper, and the initial straight section of ascending section becomes shorter.

### 4.3. Value of Ascending Section Parameters

The parameters fitted by various functions are listed in Table 5. It should be pointed out that for the cubic polynomial form recommended by Guo [21], empirical expression *A* = *E_ε_*/*f*_c_ can be used to calculate the ascending coefficient. The empirical formulas of ascending parameter *A* can be obtained by substituting the Equations (1)–(4), respectively, the expression becomes
(10)$$A_{\mathrm{sf}}=\frac{\frac{1\times 10^{5}}{2.2+(34.7/f_{\mathrm{cu}})}\varepsilon_{0}(1-0.3078\lambda_{\mathrm{sf}}-0.0777\lambda_{\mathrm{pf}})}{f_{c0}(1-0.02542\lambda_{\mathrm{pf}}+0.01306\lambda_{\mathrm{sf}})}$$
(11)$$A_{\mathrm{vf}}=\frac{\frac{1\times 10^{5}}{2.2+(34.7/f_{\mathrm{cu}})}\varepsilon_{0}(1-0.1443\lambda_{\mathrm{vf}}-0.0777\lambda_{\mathrm{pf}})}{f_{c0}(1+0.02542\lambda_{\mathrm{pf}}-0.01409\lambda_{\mathrm{vf}})}$$
In which *A*_sf_ is the ascending section parameter of specimens in PC group and *A*_pf_ is the ascending section parameter of specimens in SC group. It can be seen that the parameters of the rising section are related to the properties and content of the fibers, cubic compressive strength, and axial compressive strength of matrix. The average ratio of the results calculated by Equations (10) and (11) to the experimental results is 1.02 and the mean square deviation is 0.021.

### 4.4. Descending Section Curve Equation

At present, functions for fitting the descending section of concrete material are rare; the expressions are summarized in Table 5. In this paper, the segmented SIR model and rational fraction function suggested by Guo et al. are used. The fitting results of each formula are shown in Figure 8. It can be seen that when *x* ≤ 2, the rational fraction can reflect the falling section of the curve well, but the curves begin to separate near *x* = 2 and the residual stress section after the convergence point of the curve cannot be well reflected by rational fraction when the downward curve is *x* > 2. The result of the SIR model fitting the descend curve well. The residual stress of specimens can be reflected when the strain is large.

## 5. Conclusions and Prospect

In order to establish the compressive constitutive relationship of HFPC and obtain the influence of the composition and proportion of different fibers on the mechanical properties, several methods are presented in this article, and the following conclusions can be drawn.

(1) For the specimens mixed with hybrid fibers, the compressive strength will be reduced to varying degrees, but deformation ability will improve significantly. The higher the proportion of PP fibers in steel-PP hybrid fibers and PVA-PP specimens with fixed volume fraction, the more significant the decrease of compressive strength and modulus of elasticity.

(2) The empirical formulas for calculating the cubic compressive strength, peak stress and strain, and elasticity modulus of HFRC specimens are put forward by testing and summarizing the existing experimental data, and the applicability is wider, and the variation law of the specimens with the uniaxial compressive strength can be accurately predicted.

(3) Several formulas for calculating the stress-strain curves of HFRC specimens under uniaxial compression are put forward through experiments and verified by other measured results. The calculated values are in good agreement with the measured values. The piecewise SIR model presented in this paper has the best calculating effect. It is also feasible to use the constitutive equation suggested by Guo et al. and other methods to calculate the stress-strain curves. However, when the strain is large, it is quite different from the test results. In order to obtain a reliable downward section of the stress-strain curve under axial compression, a large number of experiments need to be carried out on the basis of the determination of standard test methods.

(4) Most of the existing formulas can reflect the rising stage of stress-strain curve well but in the declining stage, most of the formulas underestimate the residual stress of specimens. How to effectively consider the effect of hybrid fibers on the falling section curve of specimens after the peak load has been reached needs further theoretical research. At the same time, when the matrix composition changes or the fiber content exceeds the scope of this study, more experiments are needed to verify the applicability of the formulas given in this paper.

(5) In this paper, the stress-strain behavior of fiber-reinforced concrete can be described by neglecting the meso-characteristics of materials, but it is difficult to reveal the intrinsic physical mechanism of material deformation and failure. On the basis of the experiments, it is necessary to do more in-depth research in the advanced field of solid mechanics.

## Figures and Tables

**Figure 1 materials-12-02370-f001:**
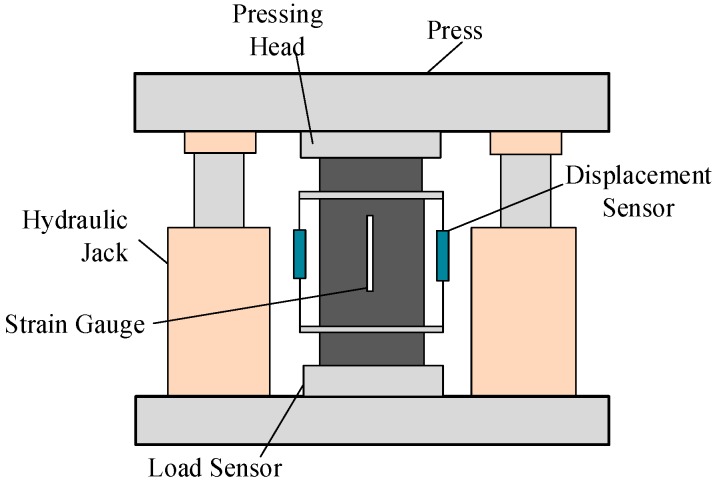
Loading Device.

**Figure 2 materials-12-02370-f002:**
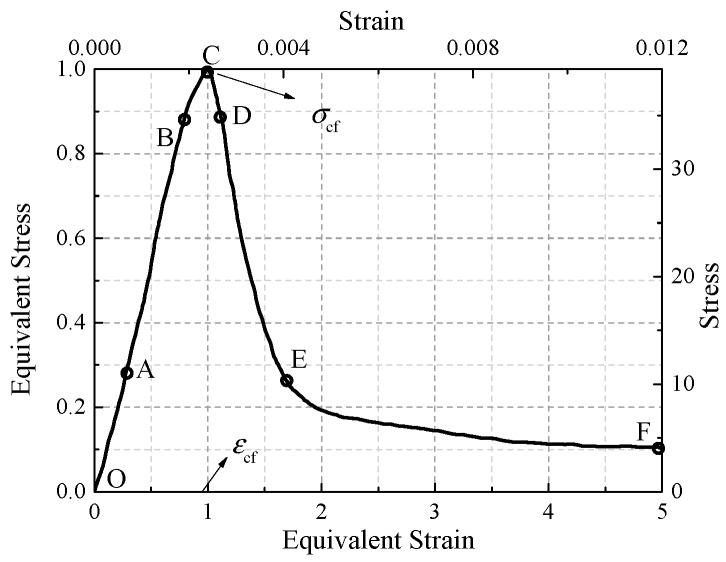
Typical stress-strain curve.

**Figure 3 materials-12-02370-f003:**
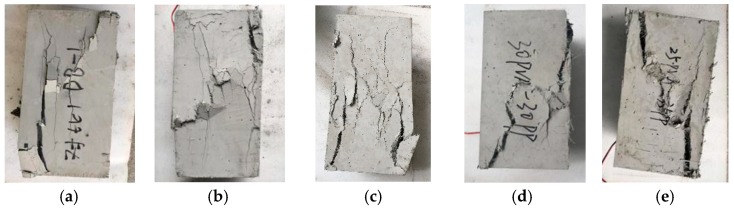
Failure mode of specimen. (**a**) DB; (**b**) PC1; (**c**) PC2; (**d**) PC3; (**e**) PC4; (**f**) PC5; (**g**) SC1; (**h**) SC2; (**i**) SC3; (**j**) SC4.

**Figure 4 materials-12-02370-f004:**
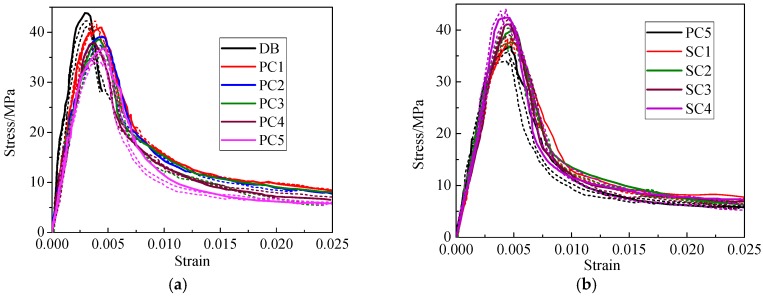
Strain-stress curves of specimen. (**a**) Specimens with PVA-PP Hybrid Fiber. (**b**) Specimens with PP-Steel Hybrid Fiber.

**Figure 5 materials-12-02370-f005:**
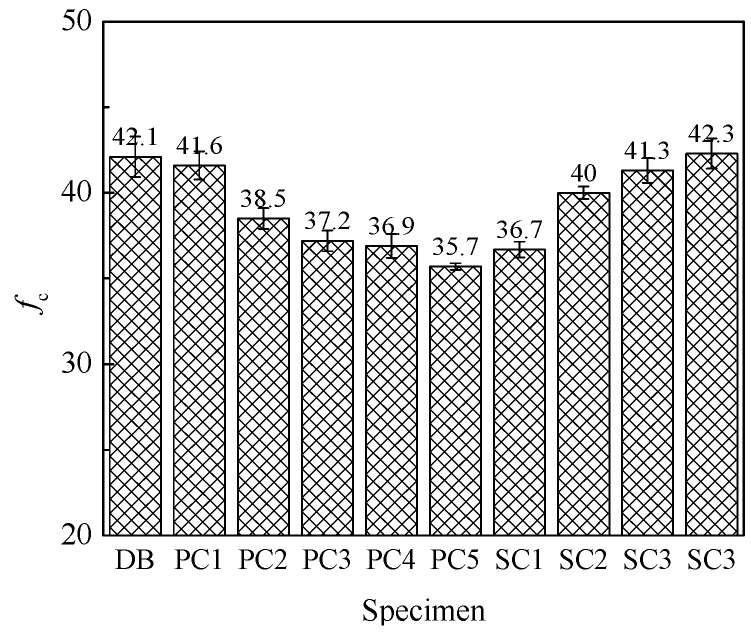
Peak stress and its error diagram.

**Figure 6 materials-12-02370-f006:**
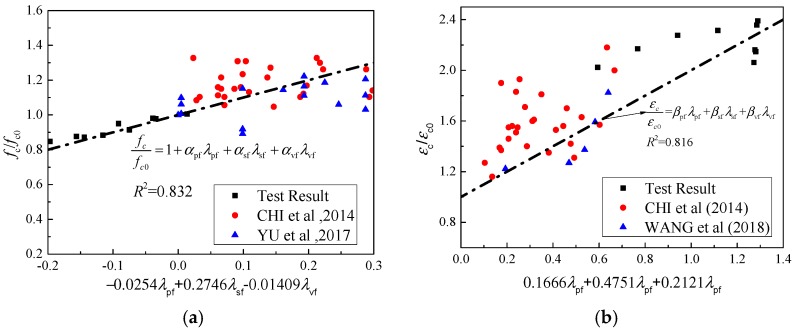
Comparison of peak and peak stress. (**a**) Comparison of peak stress. (**b**) Comparison of peak strain.

**Figure 7 materials-12-02370-f007:**
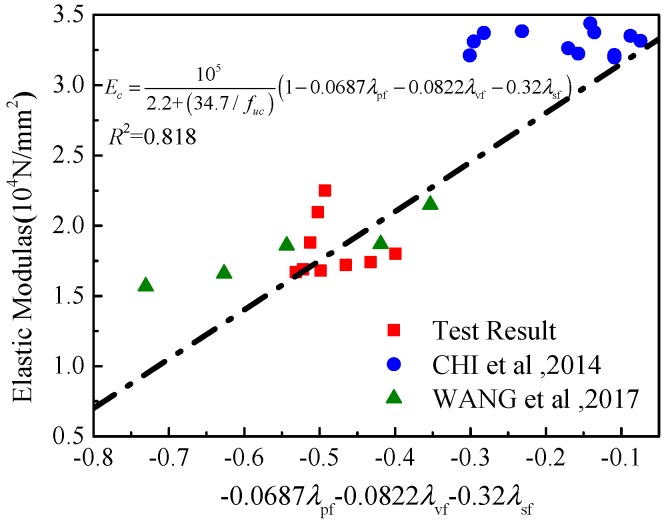
Comparison of elastic modulus.

**Figure 8 materials-12-02370-f008:**
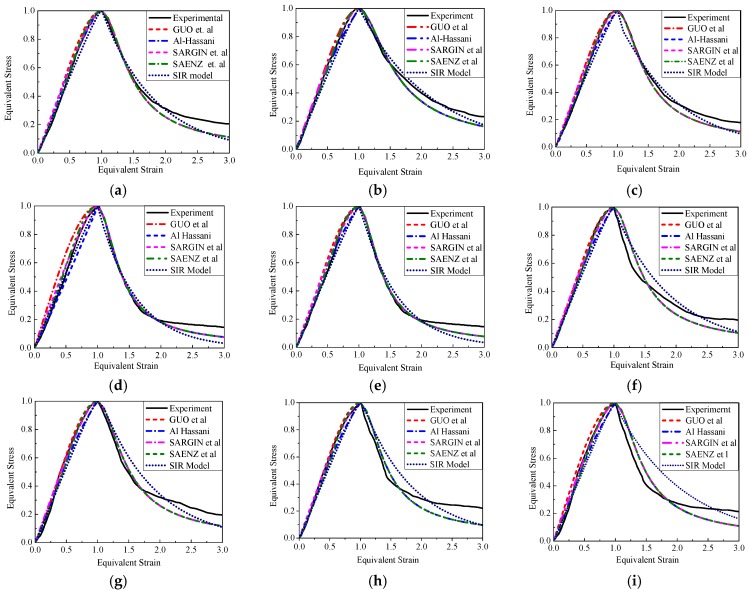
Equivalent strain-stress curve. (**a**) PC1; (**b**) PC2; (**c**) PC3; (**d**) PC4; (**e**) PC5; (**f**) SC1; (**g**) SC2; (**h**) SC3; (**i**) SC4.

**Table 1 materials-12-02370-t001:** Mix proportion of specimen.

Code	Cement kg/m^3^	Fly Ash kg/m^3^	Quartz Sand kg/m^3^	Water kg/m^3^	Water Reducing g/m^3^	PVA Fiber/%	PP Fiber/%	Steel Fiber/%
DB	475	710	415	400	35	-	-	-
PC1	475	710	415	400	35	2	-	-
PC2	475	710	415	400	35	1.5	0.5	-
PC3	475	710	415	400	35	1	1	-
PC4	475	710	415	400	35	0.5	1.5	-
PC5	475	710	415	400	35	-	2	-
SC1	475	710	415	400	35	-	-	2
SC2	475	710	415	400	35	-	0.5	1.5
SC3	475	710	415	400	35	-	1	1
SC4	475	710	415	400	35	-	1.5	0.5

**Table 2 materials-12-02370-t002:** Chemical composition of cement.

Component	SiO_2_	CaO	Al_2_O_3_	Fe_2_O_3_	MgO	SO_3_	Oher	Loss
**Mass Percent**	21.69	62.55	4.38	3.34	2.08	2.89	1.41	1.59

**Table 3 materials-12-02370-t003:** Mean value of characteristic point on σ-ε curve.

Code	*f* _cu_	*f* _c0_	*f* _f_	*ε*_c0_/%	*ε*_f_/%	*ε*_0.85_/%	*E* _0_	*E* _f0_
DB	55.6	42.1	42.1	1.80	1.80	-	2.27	-
PC1	52.8	41.2	41.8	4.25	4.34	5.03	1.79	1.78
PC2	50.0	38.5	39.1	4.57	4.64	5.32	1.73	1.72
PC3	47.9	37.2	37.5	4.78	4.77	5.48	1.67	1.68
PC4	46.1	36.9	36.2	4.86	4.80	5.45	1.71	1.76
PC5	45.2	35.7	35.1	5.02	4.97	5.59	1.66	1.63
SC1	47.6	36.7	36.8	4.95	4.92	5.50	1.68	1.62
SC2	50.6	40.0	40.2	4.51	4.73	5.51	1.87	1.90
SC3	53.6	41.3	41.1	4.54	4.42	5.23	2.12	2.15
SC4	54.1	42.3	42.4	4.33	4.24	5.11	2.21	2.22

Note: In the table *f*_f_ is the result of peak stress calculated according to Equation (1) and *ε*_f_ is the result of peak strain calculated according to the Equation (4). *ε*_0.85_ is the corresponding strain when the load-carrying capacity of the specimen decreases to 85% of the peak stress. *E*_0_ and *E*_f0_ are the results of elastic modulus measured and calculated according to the formula in this paper, respectively, in units of 10^4^ N/mm^2^.

**Table 4 materials-12-02370-t004:** Forms of uniaxial stress-strain curve.

Expression Form	Formula Source	Expression
Polynomial	Guo [20]	$y=\left\{ \begin{array}{cc} Ax+(3-3A)x^{2}+(A-2)x^{3} & 0\leq x\leq 1\\ \frac{x}{B{\left(x-1\right)}^{2}+x} & x\geq 1 \end{array}\right.$
Rational fraction	Sargin, et al. [20]	$y=\left\{ \begin{array}{cc} \frac{Ax-x^{2}}{1+(A-2)x} & 0\leq x\leq 1\\ \frac{x}{B{\left(x-1\right)}^{2}+x} & x\geq 1 \end{array}\right.$
	Al-Hassani [21]	$y=\frac{ax^{b}}{c+x^{d}}\begin{array}{cc} & x\leq 1 \end{array}$
	Saenz, et al. [20]	$y=\left\{ \begin{array}{cc} \frac{Ax}{1+(2A-3)x^{2}+(2-A)x^{3}} & 0\leq x\leq 1\\ \frac{x}{B{\left(x-1\right)}^{2}+x} & x\geq 1 \end{array}\right.$
Exponential	SIR Model [19]	$y=c(e^{ax}-e^{bx})$

**Table 5 materials-12-02370-t005:** Fitting parameters and determining coefficient test.

Stage	Source	Parameter	PC1	PC2	PC3	PC4	PC5	SC1	SC2	SC3	SC4
Ascending Section	Guo, et al. [20]	*A*	1.337	1.466	1.578	2.371	1.420	1.405	1.387	1.391	1.42
		*R* ^2^	0.987	0.991	0.986	0.989	0.981	0.981	0.979	0.984	0.977
	Al-Hassani [21]	*a*	0.021	0.038	0.087	0.071	0.032	0.098	0.087	0.0135	0.045
		*b*	1.37	1.62	1.81	1.97	2.29	1.76	1.88	1.21	1.73
		*c*	2.08	7.93	93.11	23.98	108.67	20.18	90.19	65.10	81.09
		*d*	1.45	1.98	1.99	2.01	2.34	1.97	2.09	1.43	1.92
		*R* ^2^	0.993	0.994	0.996	0.996	0.995	0.993	0.996	0.995	0.995
	Sargin, et al. [20]	*A*	1.253	1.281	1.312	1.378	1.431	1.409	1.377	1.365	1.309
		*R* ^2^	0.994	0.993	0.995	0.993	0.995	0.992	0.996	0.994	0.993
	Saenz, et al. [20]	*A*	1.094	1.117	1.181	1.277	1.295	1.247	1.234	1.212	1.197
		*R* ^2^	0.994	0.995	0.995	0.994	0.994	0.995	0.994	0.996	0.994
	SIR Model [19]	*a*	−1.92	−2.02	1.43	−1.87	−2.34	−2.91	−2.65	−2.22	−1.89
		*b*	−1.32	−1.51	1.98	−1.28	−1.99	−2.10	−2.19	−1.62	−1.21
		*c*	19.21	21.21	16.92	28.87	22.11	17.87	26.76	24.12	31.11
		*R* ^2^	0.992	0.995	0.995	0.993	0.995	0.994	0.994	0.995	0.996
Descending Section	Guo, et al. [20]	*B*	9.06	3.36	4.91	5.08	18.31	6.41	7.62	6.98	5.41
		*R* ^2^	0.968	0.980	0.974	0.961	0.959	0.946	0.942	0.934	0.971
	SIR Model [19]	*a*	−22.07	−10.8	−1.18	−2.14	−2.31	−1.06	−1.25	−1.19	−1.12
		*b*	−19.7	−22.6	−22.1	−21.4	−22.38	−21.8	−21.2	−22.1	−22.4
		*c*	−6.09	2.43	3.41	−3.31	9.80	2.55	3.34	3.10	2.98
		*R* ^2^	0.971	0.988	0.985	0.975	0.967	0.975	0.972	0.964	0.945

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
