# Peer review of "Experimental Research on Uniaxial Compression Constitutive Model of Hybrid Fiber-Reinforced Cementitious Composites"

_materials, 2019, doi:10.3390/ma12152370_

Reviewer 1 Report

The manuscript aims to establish an accurate compressive constitutive model of Hybrid Fiber Reinforced Concrete containing PVA fibers, PP fibers and steel fibers at different levels. The obtained experimental results are compared with the existing data and models.

From the analysis of the presented manuscript I found the following:

1)      Abstract: The first sentence is far too long and too complicated.

2)      Abstract: The terms PVA, PP fibers and SIR model should be written fully when firstly mentioned.

3)      Materials and Mix proportions: The chemical composition (XRF) of cement should be provided.

4)      Materials and Mix proportions: The technology of mixing and the sample preparation is missing and should be added. What were the curing conditions?

5)      Materials and Mix proportions: The w/c should be explicitly written in the text as well as the age of the studied specimens.

6)      Test Results and Analysis: The obtained results should be compared to the literature. There are two pages without any reference or a comparison whereas this study aims to compare results with existing literature.

7)      Test Results and Analysis: The description of the section 2.2 should be improved. Some sentences are difficult to follow. Generally, there is no discussion about the obtained results.

8)      Line 181: “According to the mechanical mixing law of composite materials” – the related reference is missing. The same with the line 186 “Based on test data in existed research,…” and similar statements appearing in the whole text. The mentioned “existing research” should be cited in the text, not only in figures.

9)      Figure 5 compares the specimens with the same compositions? If not, the differences should be provided.

10)   Generally, it is very difficult to follow which results are a part of the presented study or what is a general statement/comparison with the existing results.

On the basis of the specified ones, I consider that the manuscript could be published after minor changes will be made to the content of the manuscript.

Reviewer 2 Report

1. Typically fibers in concrete are used to increase the tensile strength and ductility. And there are so many existing researches have shown that the uses of fibre in concrete reduce the compressive strength. Therefore, it would be nice if the authors could use the constitutive model analysis to investigate the tensile strength, ductility and fracture toughness behaviour of concrete. Also already many literatures are available in this topic and some of them are also referred here. Therefore, the authors need to highlight the novelty of their work. Existing literatures:

·         Li, Biao, et al. "Experimental investigation on the flexural behavior of steel-polypropylene hybrid fiber reinforced concrete." Construction and Building Materials 191 (2018): 80-94.

·         Afroughsabet, Vahid, and Togay Ozbakkaloglu. "Mechanical and durability properties of high-strength concrete containing steel and polypropylene fibers." Construction and building materials 94 (2015): 73-82.

·         HUA, Yuan, Jun-ying LIAN, and Tai-quan ZHOU. "Relationship between the Mechanical Properties of Hybrid Fiber Reinforced Concrete and Length/Diameter Aspect Ratio of Hybrid Fiber [J]." Journal of Building Materials 1 (2005).

·         Gao, Danying, Han Li, and Fan Yang. "Performance of polypropylene-steel hybrid fiber reinforced concrete after being exposed to high temperature." Fuhe Cailiao Xuebao(Acta Materiae Compositae Sinica) 30.1 (2013): 187-193.

2. It is also not clear how the empirical equations are developed here. What are the factors that were consider for the development of these equations and also the limitations of these formulas are not clear.

3. How the results presented in Table 4 and in most figures can help other researchers are also not clear.

Reviewer 3 Report

The paper is aimed at experimentally investigating the uniaxial compression constitutive response of Hybrid Fiber Reinforced Cementitious Composites. A simplified model was also proposed for the same purpose. The work contains potentially nice material to be published in Materials J. (especially in the Section “Construction and Building Materials”).

However, in the Reviewer's opinion the paper needs major improvements before its approval.

The following recommendations/clarifications should be considered:

·       In section 1, after the State of the Art (SoA), the Authors should clarify what are the key novelties of this paper and the main contributions of this work beyond the current SoA. They are actually missing and/or not properly addressed.

·       Another key issue is that the authors wrote (lines 49-51): “Compressive constitutive relationship is a necessary condition of HFRC component and construction research. Its accuracy could significantly affect the design results of structures applying this material.” The Reviewer would like to remark that in compression, mostly of the international codes (e.g. Rilem model code, among others) suggest to completely avoid the fibers contribution in design, under compression stress states: thus just use the expression for unreinforced concrete. The authors need to better motivate what is the key novelty of performing a detail modelling strategy for compression of FRCC.

·       2% of volume fraction is a very high (sometimes not feasible value) fiber content in structural applications.

·       More than one repetition test (per mixture) was done for the compressive cases (3 specimens). Please plot the results by showing also the scatter of them (through histograms and error bars by showing the results of the maximum compressive strengths and by showing the grey area of variability in Figure 4). This will help to check the quality of the experimental results and the quality of the followed lab procedures. Otherwise the discussion cannot be completely believed by the Reader viewing only the mean values of the test evidences/results.

·       Table 1: please also report the fiber contents in volume fraction.

·       Please carefully read again the paper. Many typos and English errors have been found.

·       Future developments which will follow to this research paper are outlined. It should be reported at the end of the concluding section.

Author Response

Round  2

Reviewer 2 Report

Minor correction:

Title of table 2 is not correct. These are not the mix proportion.

Author Response

Thank the reviewer for correcting. In fact, Table 2 is the chemical composition table of cement. I inadvertently wrote the title incorrectly and corrected it in the revised manuscript. We apologize for this careless error.

Reviewer 3 Report

The authors improved the manuscript following the Reviewer´s indications. The paper can be accepted. 

Author Response

Reviewers'suggestions for this manuscript have greatly improved the level of this article., we thank you again for your work, we have made minor changes to the manuscript, has been marked in red font in the manuscript.